# In Vitro Characterization of Polysaccharides from Fresh Tea Leaves in Simulated Gastrointestinal Digestion and Gut Microbiome Fermentation

**DOI:** 10.3390/foods13101561

**Published:** 2024-05-16

**Authors:** Qiaoyi Zhou, Jinjing Gao, Xueyan Sun, Yicheng Liang, Minqi Ye, Dongxia Liang, Caijin Ling, Binghu Fang

**Affiliations:** 1National Reference Laboratory of Veterinary Drug Residues, College of Veterinary Medicine, South China Agricultural University, Guangzhou 510640, China; zhouqyi@foxmail.com (Q.Z.); gjjxka@163.com (J.G.); xueyansun@163.com (X.S.); 15521282309@163.com (Y.L.); a172055990@163.com (M.Y.); 2Tea Research Institute, Guangdong Academy of Agricultural Sciences, Guangdong Provincial Key Laboratory of Tea Plant Resources Innovation and Utilization, Guangzhou 510640, China; liangdx3407@163.com

**Keywords:** tea polysaccharide, simulated gastrointestinal digestion, fecal fermentation, gut microbiota

## Abstract

Tea plants have a long cultivation history in the world, but there are few studies on polysaccharides from fresh tea leaves. In this study, tea polysaccharides (TPSs) were isolated from fresh tea leaves. Then, we investigated the characteristics of TPSs during in vitro simulated digestion and fermentation; moreover, the effects of TPSs on gut microbiota were explored. The results revealed that saliva did not significantly affect TPSs’ molecular weight, monosaccharide composition, and reducing sugar content, indicating that TPSs cannot be digested in the oral cavity. However, TPSs were partially decomposed in the gastrointestinal tract after gastric and intestinal digestion, resulting in the release of a small amount of free glucose monosaccharides. Our in vitro fermentation experiments demonstrated that TPSs are degraded by gut microbiota, leading to short-chain fatty acid (SCFA) production and pH reduction. Moreover, TPSs increased the abundance of *Bacteroides*, *Lactobacillus*, and *Bifidobacterium* but reduced that of *Escherichia*, *Shigella*, and *Enterococcus*, demonstrating that TPSs can regulate the gut microbiome. In conclusion, TPSs are partially decomposed by gut microbiota, resulting in the production of SCFAs and the regulation of gut microbiota composition and function. Therefore, TPSs may be used to develop a prebiotic supplement to regulate the gut microbiome and improve host health.

## 1. Introduction

Tea is a widely consumed beverage worldwide; according to the International Tea Commission, global tea consumption reached more than 5.8 million t in 2019 [1]. However, the increase in tea consumption has also led to a significant increase in tea waste during production, including unpicked tea and trimmed tea leaves and branches, thus resulting in substantial biomass loss and environmental pressure [2]. Fresh tea leaves are rich in polyphenols, polysaccharides, lignin, and fiber, making tea a natural plant with abundant nutritional and health benefits [3]. Extracting functional components from fresh or pruned tea leaves can aid in reducing resource consumption and environmental pollution, as well as improving resource utilization efficiency and ecological benefits.

Tea contains various bioactive compounds, such as tea polyphenols (TPPs), theanine, tea polysaccharides (TPSs), and theaflavins, which contribute to the overall health of humans. Studies thus far have primarily focused on understanding the physicochemical properties and biological activities of the low-molecular-weight (MW) components in tea, particularly catechins and theanine [4,5]. TPSs, with relatively high MWs, are bioactive and demonstrate a significant potential for development into nutritional supplements; therefore, they warrant further exploration. In recent years, nutritional and biochemical scientists have increasingly recognized the importance of polysaccharides—natural high-MW polymers connected by glycosidic bonds formed by aldose or ketose. Plant polysaccharides exhibit various biological activities, including antiviral [6], anticancer [7], antioxidant [8], hypoglycemic [9], and immunoregulatory [10,11]. Consequently, they have become a research focus in the fields of food science, natural drugs, biochemistry, and life sciences globally; moreover, they have been widely used in the fields of human health and clinical medicine. Similarly, TPSs have gained considerable attention. Relevant research thus far has focused on TPS extraction, purification, physical and chemical properties, and pharmacological effects. The research objects of tea polysaccharides are mainly focused on different types of finished tea, while there are few studies on the structural information and mechanism of polysaccharides extracted from fresh tea leaves.

In recent years, significant research in the fields of microbiology, medicine, and genetics has focused on gut microbiota and demonstrated that gut microbiota plays a crucial role in host health, nutrition, metabolism, and immune homeostasis. Natural polysaccharides can improve the intestinal microenvironment and impact host health through various mechanisms. These mechanisms include improving gut microbiota composition, enhancing intestinal barrier function, increasing antioxidant activity, promoting short-chain fatty acid (SCFA) production, and reducing proinflammatory mediator levels [12,13,14]. Furthermore, plant polysaccharides can serve as substrates for gut microbiota fermentation, influencing their structure and metabolism [15]. This, in turn, affects glucose metabolism, intestinal immune system development, lymphocyte immune response, and, ultimately, host health improvement dose-dependently [16].

Plant polysaccharides are now considered to have potential prebiotic effects, selectively stimulating the growth of some probiotics and thereby restoring the homeostasis of the intestinal bacterial community [17]. Studies have found that ginseng polysaccharides improve the intestinal metabolism of some primary ginsenosides (ginsenosides Re and Rc) and secondary ginsenosides (20-(S) Rg2, Rd, and 20-(S) Rg3) after entering the gastrointestinal tract and, as prebiotics, restore the balance of the intestinal ecosystem, particularly by promoting the growth of the probiotic bacteria *Lactobacillus* and *Bacteroides*. Ginseng polysaccharides also have indirect gut microbiota-mediated treatment effects. The research on plant polysaccharide–gut microbiome interactions is in its initial stages; nevertheless, the relevant findings have improved the current understanding of the scientific basis of plant polysaccharides. As such, gut microbiota may be an important target for pharmacological treatment using plant polysaccharides. Therefore, the exploration of the interactions between gut microbiota and plant polysaccharides may provide novel directions for the analysis of the mechanisms underlying the effects of plant polysaccharides in animals [18].

Compared with in vivo experimentation, the use of in vitro digestion and gut microbiota fermentation models is associated with a lower cost, simpler operation, and higher reproducibility. They can simulate the digestion process and demonstrate the impact of nutrients on human health in vitro [19]. Therefore, these models have been widely used to evaluate the digestion and fermentation characteristics of functional components [20]. Studies have demonstrated that TPSs are not destroyed during orogastrointestinal digestion and thus can reach the large intestine smoothly [21,22]. TPSs can also be used by gut microbiota to synthesize beneficial nutrients, which are transported throughout the body via the circulatory system and regulate health. However, the relevant research so far has mainly focused on TPSs, such as Fuzhuan [23], Liubao [24], and Taiping Houkui TPSs [25], extracted from finished tea or tea dregs. Few studies have directly extracted polysaccharides from fresh tea leaves or purified TPS components. The monosaccharide composition and content of TPSs obtained from different tea raw materials or tea preparation methods are different; therefore, these TPSs may demonstrate differences in biological activities [26,27]. Therefore, the potential digestion mechanisms of action and regulatory effects of TPSs on gut microbiota warrant exploration.

In this study, we investigated changes in TPSs and their purified component (TPS-D1N1) in an in vitro digestion and fecal fermentation model. We also examined the effects of TPSs on gut microbiome composition, diversity, and SCFA production. The current findings may provide fundamental data for the use of TPSs as functional components in nutrition, as well as for the development of newer drugs.

## 2. Materials and Methods

### 2.1. Materials and Reagents

Fresh tea leaves were obtained from Zilong Agricultural Development Co., Ltd. (Heyuan, China). Standards of monosaccharides, namely mannitol (Man), ribose (Rib), rhamnose (Rha), glucuronic acid (Glc-UA), galacturonic acid (Gal-UA), glucose (Glc), galactose (Gal), xylose (Xyl), arabinose (Ara), fucose (Fuc), 3-methyl-1-phenyl-2-pyrazolin-5-one (PMP), and trifluoroacetic acid (TFA), were purchased from Sigma-Aldrich (St. Louis, MO, USA). SCFA standards, namely acetic acid, propionic acid, butyric acid, isobutyric acid, valeric acid, isovaleric acid, hexanoic acid, 1-ethyl-(3-dimethylaminopropyl) carbodiimide hydrochloride (1-EDC·HCl), and 2-nitrophenyl hydrazine hydrochloride (2-NPH·HCl), were purchased from Aladdin Reagents (Shanghai, China). Fructooligosaccharides (FOSs) were purchased from Quantum Hi-Tech Biological (Guangdong, China). Artificial saliva, artificial gastric juice, and artificial intestinal juice were provided by Shanghai Yuanye Biological Technology (Shanghai, China). Artificial saliva (ISO/TR10271 [28], neutral) was mainly composed of NaCl, KCl, CaCl_2_, NaH_2_PO_4_, urea, Na_2_S, etc. Artificial gastric fluid (pH 1.5) was mainly composed of sodium chloride, dilute acid, pepsin, etc. Artificial small bowel fluid (pH 6.8) was mainly composed of phosphate, pancreatic enzymes, etc. All other chemical reagents were analytical grade. Balb/c mice were used in this study under the approval of the Experimental Animal Ethics Committee of South China Agricultural University (Approval No. 2023b160).

### 2.2. TPS Extraction and Preparation

With fresh tea leaves as the raw material, TPSs were extracted using the aqueous alcoholic precision method reported by Zheng et al. [29] but with slight modifications. In brief, naturally air-dried, ground, and sieved fresh tea leaves were treated twice with absolute ethanol at a ratio of 1:25 (*w*/*v*) at 60 °C to remove alcohol-soluble substances such as small molecules, oligosaccharides, and pigments. Subsequently, the tea residue was extracted with water at a ratio of 1:25 (*w*/*v*) twice in a water bath at 60 °C. The resulting extract was then concentrated to 10% of the original volume by using vacuum rotary steam, followed by precipitation with anhydrous ethanol at 4 °C for 12 h. After centrifugation, the samples were dissolved in water and freeze-dried to obtain a crude TPS extract. The TPS extract was purified further by using the Sevage method to remove proteins, followed by the petroleum ether method to remove fats and the use of AB 8 macroporous resin and a dialysis bag (3000 Da) to eliminate impurities such as small molecules; the obtained crude extract was labeled TPSs. These TPSs were then subjected to purification by using a DEAE Seplife FF weak anion-exchange column (Xi’an, China) and a Sephacryl S-400 HR gel separation column (GE Healthcare, Uppsala, Sweden). Finally, dialysis was performed using a dialysis bag to remove salts, and after vacuum freeze-drying, a purified TPS fraction was obtained and labeled TPS-D1N1.

### 2.3. TPS Characterization

#### 2.3.1. Total Carbohydrate and Protein Content Determination

We determined the total carbohydrate and protein contents by using the phenol sulfuric acid [30] and Bradford [31], respectively.

#### 2.3.2. Monosaccharide Composition Determination

TPS monosaccharide composition was determined by precolumn derivatization high-performance liquid chromatography (HPLC) as described previously but with slight modifications [32]. In brief, TPS samples were mixed with 1 mL of 4 M TFA and reacted in a water bath at 100 °C for 4 h. After the completion of the reaction, 2 mL of methanol was added, and the mixture was dried. This process was repeated five times to remove TFA completely. Then, 0.5 mL of ultrapure water was added to the TPS samples for hydrolysis. Our TPS samples or monosaccharide standards were thoroughly mixed with 0.5 mL of 0.6 M NaOH and 1 mL of 0.5 M PMP methanol. The reaction was performed in a water bath at 70 °C for 100 min. After the completion of the reaction, 1 mL of 0.3 M HCl was added to neutralize the mixture, which was then evaporated to dryness under reduced pressure. The dried mixture was mixed with 2 mL of purified water and then vibrated until completely dissolved. Subsequently, 2 mL of chloroform was added for extraction; this process was repeated five times in the upper water phase. The resulting membrane filtrate was used for HPLC analysis with these parameters: column, ZORBAX Eclipse XDB-C18 column (4.6 × 250 mm^2^); column temperature, 30 °C; detection wavelength, 245 nm; mobile phase, 0.1 M phosphate-buffered saline (pH 6.7) and acetonitrile (83:17, *v*/*v*); flow rate, 1 mL/min; and injection volume, 20 μL.

#### 2.3.3. MW Determination

MW was determined on a gel chromatography differential multiangle laser light scattering system [33] equipped with UltiMate 3000 (Thermo Fisher, Waltham, MA, USA), Optilab T-Rex (Wyatt, CA, USA), and Dawn Helios Ⅱ (Wyatt) as the liquid system, differential detector, and laser light scattering detector, respectively. We used OHpak SB-805 HQ (300 × 8 mm^2^, Showa Denko, Tokyo, Japan) and SB-803 HQ (300 × 8 mm^2^, Showa Denko, Tokyo, Japan) in series as the gel exclusion chromatographic columns, with the column temperature and injection volume of 45 °C and 100 μL, respectively. The mobile phase consisted of 0.02% NaN_3_ and 0.1 M NaNO_3_, with a flow rate and elution gradient of 0.6 mL/min and 75 min, respectively.

### 2.4. In Vitro Simulation of Orogastrointestinal Digestion

#### 2.4.1. In Vitro Simulation of Oral Digestion

The digestion process of TPSs was modified based on the method reported by [34,35,36]. To simulate the oral digestion process, TPSs or TPS-D1N1 (4 mg/mL) and artificial saliva were mixed at a 1:1 ratio in a centrifuge tube and stirred in a shaker at 100 rpm at 37 °C. Next, 2 mL of the digested samples were withdrawn at 0, 0.25, and 0.5 h and heated at 100 °C for 5 min to deactivate the salivary enzymes.

#### 2.4.2. In Vitro Simulation of Gastric Digestion

After 0.5 h of salivary digestion, the solution was mixed with the artificial gastric juice at a 1:1 ratio (*v*/*v*). The final pH of the mixture was adjusted to the typical gastric pH of 2.0 with 0.1 M HCl. This was followed by stirring in a shaker box at 100 rpm at 37 °C for 6 h. At 0, 1, 2, 4, and 6 h, 2 mL of the digested samples were withdrawn and heated at 100 °C for 5 min to deactivate the gastric enzymes.

#### 2.4.3. In Vitro Simulation of Small Intestinal Digestion

After 6 h of gastric digestion, the pH of the mixture was adjusted to the gut pH of 7.0 using 1 M NaHCO_3_. The mixture was then combined with the artificial small intestinal juice at a 10:3 ratio (*v*/*v*) and stirred in a shaker at 100 rpm at 37 °C for 6 h. At 0, 1, 2, 4, and 6 h, 2 mL of the digested samples were taken and heated at 100 °C for 5 min to deactivate the intestinal enzymes.

The MW, monosaccharide composition, and reducing sugar content of all samples withdrawn after simulated orogastrointestinal digestion were determined. Each test was repeated three times.

### 2.5. In Vitro TPS Fermentation

In vitro fermentation was performed using a previously reported method to collect fecal samples into sterile anaerobic tubes [37]. All fecal samples were mixed with 0.1 M phosphate-buffered saline (pH 7.1) to acquire a 10% (*w*/*v*) fecal slurry. The mixture was homogenized thoroughly and then centrifuged at 3000 rpm at 4 °C for 5 min. The supernatant was collected.

The basal nutrient medium was finetuned according to the method reported by [38]. In brief, we dissolved yeast extract (2.0 g), peptone (2.0 g), NaCl (0.1 g), K_2_HPO_4_ (0.04 g), KH_2_PO_4_ (0.01 g), CaCl_2_·2H_2_O (0.01 g), MgSO_4_·7H_2_O (0.01 g), NaHCO_3_ (2.0 g), heme (0.02 g), L-cysteine (0.5 g), bile salts (0.5 g), tween 80 (2.0 mL), 1% (*w*/*v*) resazurin solution (1.0 mL), and vitamin K_1_ (10.0 μL) in 1 L of ultrapure water and adjusted the pH to 7.0 with 0.1 M HCl. This medium was then sterilized at 121 °C for 20 min.

FOSs, TPSs, or TPS-D1N1 were used as the carbon source in this medium in the test groups, and basal nutrient medium without a carbon source was used in the control group. Next, 1.0 mL of the 10% fecal slurry was mixed with 9.0 mL of basal nutrient medium containing 100.0 mg of the carbon source, followed by anaerobic incubation at 37 °C for 6, 12, 24, or 48 h. After fermentation, the samples were withdrawn, inactivated in a boiling water bath for 5 min, and centrifuged at 5000 rpm for 10 min. The supernatant was used to determine carbohydrate consumption, pH, and SCFAs, and the precipitate (collected at 48 h after fermentation) was stored at −80 °C until it was used for microbiological analysis.

### 2.6. Determination of SCFA Production during Fermentation

SCFA production was determined as reported previously but with some modifications [39]. Fermentation broth (100 μL) was mixed thoroughly with 200 μL of 200 mmol/L 2-NPH·HCl, 200 μL of 120 mmol/L 1-EDC·HCl, and 200 μL of 6% pyridine and allowed to react in a water bath at 60 °C for 20 min. After derivatization, 100 μL of 15% KOH was added to terminate the reaction. The mixture was then acidized by adding 2 mL of 42.5% phosphoric acid, followed by extraction with 4 mL of diethyl ether twice. The organic phase was collected and dried under reduced pressure. Next, the residue was dissolved in 200 μL of methanol and analyzed on a ZORBAX Eclipse XDB-C18 column (4.6 × 250 mm^2^; Agilent, Santa Clara, CA, USA), with a mobile phase consisting of acetonitrile and water (pH 3.0). The elution was carried out using a gradient program, and the column temperature was maintained at 40 °C. UV detection was performed at 230 nm, and the SCFA content was calculated using a standard calibration curve.

### 2.7. DNA Extraction and 16S rDNA Gene Sequencing

After the in vitro fermentation of fecal samples, total bacterial DNA was extracted using the HiPure Stool DNA Kit (model D3141; Guangzhou Meiji Biotechnology, Guangzhou, China), according to the manufacturer’s instructions. We then amplified the 16S rDNA V3–V4 region from the extracted DNA through a polymerase chain reaction using the 341F (CCTACGGGNGGCWGCAG) and 806R (GGACTACHVGGGTATCTAAT) primers. The resulting amplicons were purified, connected to sequencing adapters to create a sequencing library, and subsequently sequenced on Illumina PE250. Finally, the sequencing reads and operational taxonomic units (OTUs) were analyzed.

### 2.8. Statistical Analyses

All experiments were conducted in triplicate; the data are presented as means ± standard deviations (SDs). One-way analysis of variance was performed using SPSS (version 22), and the statistical difference was compared using the Tukey test. A *p* value of <0.05 was considered to indicate statistical significance.

## 3. Results and Discussion

### 3.1. Basic TPS Components

As listed in Table 1, the TPS and TPS-D1N1 carbohydrate contents were 68.89% and 70.58%, respectively, confirming that polysaccharides are the main components of both TPSs and TPS-D1N1. TPSs and TPS-D1N1 were noted to contain the following monosaccharides: Man, Rib, Rha, Glc-UA, Gal-UA, Glu, Gal, Xyl, Ara, and Fuc; this indicated that TPSs and TPS-D1N1 are typical heteropolysaccharides with a high sugar content. As displayed in Table 1, the MWs of TPSs and TPS-D1N1 are 478.75 and 289.10 kDa, respectively.

### 3.2. Effects of In Vitro Simulated Digestion on TPS Changes

#### 3.2.1. Changes in TPS MW and Monosaccharides

To determine whether in vitro simulated orogastrointestinal digestion contributes to the degradation of TPSs and TPS-D1N1, we measured the MWs of polysaccharides and monosaccharides during in vitro simulated orogastrointestinal digestion. As shown in Figure 1 and Figure 2, the TPS curves for salivary and gastric digestion almost overlapped at different time points. The results confirmed that TPS MWs do not change significantly during salivary and gastric digestion (Figure 1A,B); in other words, the main TPS structure remains relatively stable throughout salivary and gastric digestion. Similar results have been noted previously for Hypsizygus marmoreus polysaccharides [40].

The TPS-D1N1 curves of salivary digestion almost overlapped at different time points (Figure 1D–F), and no change was noted in the MWs. In addition, monosaccharides were not detected after salivary digestion, indicating that the saliva did not degrade TPS-D1N1 (Figure 2A,D). During gastric juice digestion, the TPS-D1N1 peak slightly shifted to the right, indicating an extended gastric residence time after digestion. After 6 h of gastric digestion, the MW of TPS-D1N1 significantly decreased from 3784.501 to 2201.828 kDa. 

Previous studies have shown that polysaccharides can be partially degraded under acidic conditions during gastrointestinal digestion [41]. Therefore, we hypothesized that the decrease in Mw of TPS-D1N1 might be related to the acidic environment of the stomach, leading to glycosidic bond breakage in the polymer chain [42]. The average MW of TPSs continued to decrease after 2, 4, and 6 h of intestinal digestion, and gastrointestinal digestion led to the release of a small amount of glucose from TPS and TPS-D1N1 (Figure 2B,C,E,F). Here, polysaccharide degradation may be attributable to the low pH and glucose hydrolysis on the branching chain by the pancreatic enzymes [42]. 

#### 3.2.2. Changes in Reducing Sugar Content

Polysaccharide degradation can be assessed by measuring the reduction in the sugar content. Moreover, an increase in the reducing sugar content indicates a reduction in MW and the breaking of glycosidic bonds in polysaccharides [43]. As presented in Table 2, the TPS reducing sugar content remained unchanged during salivary digestion, suggesting that TPS content was not degraded. This finding is consistent with the MW and monosaccharide detection results. However, the TPS reducing sugar content demonstrated a significant increase from 127 to 182 μg/mL over 6 h of gastric digestion (*p* < 0.05) and from 200.69 to 382.64 μg/mL over 6 h of intestinal digestion (*p* < 0.05). Similar changes were observed for TPS-D1N1. Taken together, these results indicated that gastrointestinal fluids but not saliva can partially degrade TPS and TPS-D1N1. Studies have demonstrated that polysaccharides can be degraded under acidic conditions, and digestive enzymes in the small intestine enable glycosidic bond cleavage, leading to the release of some oligosaccharide monomers from the polysaccharides. In this study, the results of monosaccharide composition also showed that a small amount of Glc was produced after gastric and intestinal digestion, which further proved that the increase in reducing sugar content may be related to the cleavage of glycosidic bonds. Therefore, an increase in the reducing sugar content may be related to the acidic environment of the gastric juice and the digestive enzymes in the intestinal juice [42]. For instance, the MW stability of *Ganoderma lucidum* and mulberry polysaccharides has been noted to decrease during in vitro simulated gastrointestinal digestion, along with a significant increase in the reducing sugar contents, thus indicating the occurrence of polysaccharide degradation [44,45]. However, some studies have indicated that no significant change occurs in the MWs of aloe [46] and pumpkin [47] polysaccharides during gastric and intestinal digestion. Moreover, digestion with artificial gastric and small intestinal juices was noted to not lead to a change in *Hydrangea* polysaccharide MW or to the release of small-molecule free monosaccharides; nevertheless, a small amount of reducing sugar is generated [48]. Furthermore, under simulated orogastrointestinal digestion, *Gracilaria* polysaccharide MW did not change but led to reducing sugar content stability [49]. As such, different polysaccharide types demonstrate different digestive characteristics, possibly because of differences in the polysaccharides’ glycosidic bond types, MWs, and monosaccharide compositions.

### 3.3. Effects of In Vitro Fermentation of Indigestible TPSs

#### 3.3.1. Changes in Total Carbohydrates

Polysaccharides can be used as a carbon source by gut microbiota and produce beneficial metabolites [50,51]. As illustrated in Figure 3A, during the fermentation process, the consumption of carbohydrates in the TPS, TPS-D1N1, and FOS groups increased gradually; after 48 h of fermentation, carbohydrate consumption was 42.78%, 55.04%, and 24.42% in the TPS, TPS-D1N1, and FOS groups, respectively. Therefore, TPSs and TPS-D1N1 can be used as carbon sources for gut microbiome fermentation. Moreover, gut microbiota demonstrated differences in their ability to ferment polysaccharides with different structures and properties. Notably, the residual carbohydrate consumption remained lower in the FOS group than in the TPS and TPS-D1N1 groups, indicating the presence of differences in the carbohydrate utilization rate of gut microbiota among the groups. These results confirmed that TPS and TPS-D1N1 can be degraded and used by gut microbiota. The ability of gut microbiota to use polysaccharides may be closely related to the polysaccharides’ structure, including their chain lengths, monosaccharide compositions, MWs, glycosidic bond types, and carbohydrate polymerization degrees [52,53].

#### 3.3.2. Changes in pH

pH can be considered a crucial indicator of polysaccharide fermentation in the intestines. In general, the growth of other microorganisms in the gut microbial community is inhibited by nutritional competition and pH changes due to metabolite production [54]. The pH changes noted in the current study are displayed in Figure 3B. At the initial stage of in vitro fermentation, no significant difference was noted in pH between the blank and experimental groups (*p* < 0.05). During the fermentation process, the pH of the blank group at different time points was significantly higher than that of the TPS, TPS-D1N1, and FOS groups. After 48 h of fermentation, the pH of the FOS group decreased continually from 8.04 to 5.09, which is consistent with previous reports [55]. The pH change demonstrated a similar trend in the TPS, TPS-D1N1, and FOS groups; nevertheless, the pH of the TPS and TPS-D1N1 groups demonstrated a greater downward trend: from 7.96 to 4.05 and from 8.06 to 4.41, respectively (both *p* < 0.05). Several studies have reported similar observations of pH change [49,56]. For instance, appropriately reducing the intestinal pH can affect the production of SCFAs, promote the reproduction of some beneficial bacteria, and inhibit the growth of some pathogenic bacteria [57]. In the current study, the reduction in the intestinal pH due to TPS fermentation may be beneficial to host health; for instance, low pH reduces the formation of toxic compounds, such as ammonia, amines, and phenolic compounds, from the degradation of peptides [58]. 

#### 3.3.3. Changes in SCFAs

SCFAs are the main metabolites of carbohydrates fermented by gut microbiota; they typically include acetates, propionates, and butyrates, all of which play a major role in protecting host health and maintaining the normal physiological function of the human gut [59,60]. Bacterial fermentation of polysaccharides leads to the production of acid fermentation end products, mainly lactic acid and SCFAs, which reduce the colon pH and then affect microbial community composition in the gastrointestinal tract [54]. As shown in Figure 3C, in each group, the fermentation products were mainly acetic acid, propionic acid, and n-butyric acid; moreover, the SCFA contents differed significantly among the groups (all *p* < 0.05). After 48 h of fermentation, the total SCFA content increased from 0.61 to 22.43 mM in the TPS group and from 0.38 to 17.85 mM in the TPS-D1N1 group; this increase was significantly higher than noted in the blank (8.17 mM) and FOS (13.59 mM) groups (all *p* < 0.05). The total SCFA content increased steadily within 48 h of fermentation. This further illustrates that TPSs and TPS-D1N1 can be used as carbon sources by gut microbiota to produce SCFAs, thereby reducing the pH and thus affecting the microbial community composition.

#### 3.3.4. Changes in Gut Microbiome Diversity

We also used high-throughput 16S rDNA sequencing to investigate the effects of TPSs on the composition of gut microbiota. We obtained a total of 831,810 valid reads from all four groups (every three replicates), with an average of 69,317 ± 11,314 reads per sample. Next, we assessed the gut microbiome richness and diversity through alpha-diversity analysis (Figure 4). The gut microbiome diversity and richness represented by different indexes demonstrate some differences. For different statistical analysis indexes, the calculation methods are completely different; therefore, a completely inconsistent trend change is normal. We used the Simpson indexes as the community diversity indexes and Chaol and ACE as the community richness indexes. Compared with the blank group, the TPS and TPS-D1N1 groups did not demonstrate significant changes in the OTU numbers (Sobs). The TPS and TPS-D1N1 groups demonstrated no significant differences in gut microbiota richness indexes (Chao1 and ACE; all *p* > 0.05). In addition, compared with the blank group, the TPS-D1N1 group demonstrated a significant increase in the Simpson indexes and, therefore, higher gut microbiota diversity. In the TPS group, the Shannon index decreased significantly, but the Simpson index demonstrated no significant difference. This result indicated that the microbial diversity in the TPS group demonstrated a downward trend compared with the blank group. Species richness, as evaluated by the Shannon index, decreased in response to TPS treatment, consistent with previous reports [61], and this can likely be explained by the competitive role of dominant microbiota [62]. 

To evaluate the beta diversity of the gut microbiota, we used principal coordinate analysis (PCoA) and cluster analysis based on the Bray–Curtis distance to describe the correlation of microbial composition and structure based on OTU levels among all samples in different groups. As shown in Figure 5A,B, PCoA based on weighted UniFrac distance demonstrated that the first principal component (PCo1) and second principal component (PCo2) contributed 80.94% and 15.05% of the changes, respectively. Moreover, the samples within the blank, FOS, TPS, and TPS-D1N1 groups were extremely close and well clustered into one group. Furthermore, the distributions among the FOS, TPS, and TPS-D1N1 groups were clearly separated and at a long distance from the blank group at the PCo1 level, indicating the differences in gut microbiome responses to the three carbohydrate resources. In the clustering tree analysis of the beta diversity distance matrix (Figure 5C), different groups were clearly separated; moreover, the clusters between samples in the same group were highly concentrated, and their similarities in the clustering tree were consistent with the PCoA plot. These consistent results indicated that samples from different carbohydrate resources affect microbial communities differently, suggesting the specificity of bacterial distribution. Therefore, among all three experimental groups, the TPS and TPS-D1N1 groups had similar gut microbiome structures; however, these groups had gut microbiome structures significantly different from those of the blank and FOS groups.

#### 3.3.5. Changes in Gut Microbiome Composition in Fermentation Culture

Figure 6A,B presents the microbial community structure analysis results at the phylum level. Proteobacteria, Bacteroidota, and Firmicutes were the dominant bacterial phyla in all samples. This is consistent with the results reported previously: >90% of the gut microbiome comprises Bacteroidota and Firmicutes bacteria [63]. In the current study, over 48 h of fermentation, the gut microbiome composition in the TPS, TPS-D1N1, and FOS groups changed significantly. Compared with the blank group, the TPS and TPS-D1N1 groups demonstrated significantly higher levels of Bacteroidota (*p* < 0.05) but significantly lower levels of Proteobacteria (*p* < 0.05). In addition, the results showed that the abundance of TPS group had the highest abundance of Bacteroidota among all groups, suggesting that TPS may be used by *Bacteroides*, thus resulting in an increase in bacterial abundance. Studies have indicated that Bacteroidota bacteria can shape the enteric landscape by modulating host immunity and degrading diet- and host-derived glycans [64]. In addition, Bacteroidota bacteria contain various carbohydrate-active enzymes that specifically degrade nondigestible polysaccharides to produce acetates and propionates, which can be used by other gut microbiota [65]. Overall, these results demonstrate that TPS treatment can efficiently modulate the gut microbiota, with enrichment for Bacteroidota bacteria. Proteobacteria is a phylum containing pathogenic bacteria, mainly including *Escherichia coli*, *Shigella*, *Salmonella*, and *Campylobacter*, all of which can cause mild inflammation and even chronic colitis. An increase in the number of Proteobacteria bacteria may cause inflammatory reactions; therefore, these bacteria are potential diagnostic markers of biological disorders and disease risk [42]. In this study, Bacteroidota bacteria were the predominant bacteria, which degraded or used TPSs and TPS-D1N1 in the fermentation broth. The increase in Bacteroidota bacteria indicates that TPSs can be degraded and used by some Bacteroidota bacteria, promoting SCFA production in the intestines. This result is similar to that previously reported for Fuzhuan brick TPSs [24] and grapeseed polysaccharides [66]. In the current study, the relative abundance of Proteobacteria in the TPS and TPS-D1N1 groups (39.77% and 38.10%, respectively) was lower than that in the blank group; this result indicated that exposure to TPSs and TPS-D1N1 can reduce the number of endotoxin-producing *Proteus*, which has a positive effect on maintaining intestinal barrier health [67]. Compared with the blank group, the FOS group demonstrated a significant increase in the relative proportion of Proteobacteria; this effect may be related to the use of a low-MW carbon source for Proteobacteria growth.

Firmicutes is the largest and most abundant bacterial phylum. Some Firmicutes bacteria in the large intestine can produce butyrates. These bacteria can ferment indigestible polysaccharides, increasing their cell count and butyric acid production in the colon. This process aids in improving gut function, benefiting the host [68]. However, it has been reported that an increased ratio of Firmicutes to Bacteroidota may contribute to obesity because it is associated with increased colonic fermentation and energy extraction of SCFAs [69]. Moreover, the presence of Firmicutes was found to be associated with diets high in sugar and fat in mice [70]. Therefore, we speculate that there is a close relationship between the increased content of Firmicutes and the intervention of TPSs. In this study, the Firmicutes abundance in the TPS-D1N1 and FOS groups was significantly higher than that in the blank group (*p* < 0.05). Therefore, Firmicutes was a major bacterial phylum in the TPS-D1N1 and FOS groups, possibly because of the low pH generated after 48 h of fermentation. In addition, the relative abundance of Actinobacteria in the TPS, TPS-D1N1, and FOS groups was significantly higher than that in the blank group (*p* < 0.05). The increase in Actinobacteria may mainly be attributable to the increase in the number of *Bifidobacterium*. *Bifidobacterium* spp. are considered one of the important probiotics in the human intestinal tract; they can control the level of serum cholesterol, prevent intestinal disorders, and regulate the immune system [71]. Therefore, FOSs, TPSs, and TPS-D1N1 can regulate the gut microbiome composition to change the potential of energy metabolism.

Figure 6C–G illustrates the distribution of gut bacteria at the genus level in each group after 48 h of in vitro fecal fermentation. The blank group was mainly composed of *Escherichia*–*Shigella* (53.02%), *Bacteroides* (13.22%), and *Enterococcus* (7.25%). Compared with the blank group, the TPS and TPS-D1N1 groups demonstrated a significant increase in the numbers of beneficial microorganisms such as *Bacteroides*, *Lactobacillus*, and *Bifidobacterium* (*p* < 0.05) but a significant decrease in those of the pathogenic *Escherichia–Shigella* and *Enterococcus* (*p* < 0.05). In hosts, *Bifidobacterium* and *Lactobacillus* inhibit growth and infection by pathogenic bacteria, synthesize essential vitamins, promote mineral absorption, produce intestinal peristalsis-stimulating organic acids (e.g., acetates, propionates, butyrate, and lactates), promote defecation, enhance muscle strength and function, reduce oxidative stress and inflammation in peripheral tissues, improve intestinal barrier function, and stimulate the immune system to improve disease resistance [72]. *Escherichia–Shigella* causes gastrointestinal diseases. Although they form the normal intestinal flora, *Enterococcus* spp. are the main causative agents of urinary tract, skin, and soft tissue infections [73].

Taken together, these results indicated that TPSs and TPS-D1N1 can inhibit the growth of pathogenic microorganisms and promote that of beneficial microorganisms.

## 4. Conclusions

In this study, we simulated upper gastrointestinal digestion and fecal fermentation of TPSs in vitro to investigate the impact of TPSs on the digestive properties and gut microbiota. The results indicated that salivary amylase does not affect the TPSs’ MW and total sugar content. However, after undergoing gastric and intestinal digestion, the TPSs demonstrated a decrease in MW; the resulting gastrointestinal fluid demonstrated the presence of free monosaccharides. This result suggested that monosaccharides are produced during the gastrointestinal digestion of TPSs. TPS and TPS-D1N1 fermentation by gut microbiota led to a significant decrease in the total carbohydrate content and pH but a considerable increase in SCFA production. Furthermore, TPSs increased the abundance of the beneficial *Bacteroides*, *Lactobacillus*, and *Bifidobacterium*, whereas they reduced that of the pathogenic *Escherichia*, *Shigella*, and *Enterococcus*. Our findings demonstrated that gut microbiota can break down TPSs to generate SCFAs and regulate the gut microbiome composition and function. Therefore, TPSs hold potential as a functional product for promoting gut health; however, further research evaluating their in vivo effects is warranted.

## Figures and Tables

**Figure 1 foods-13-01561-f001:**
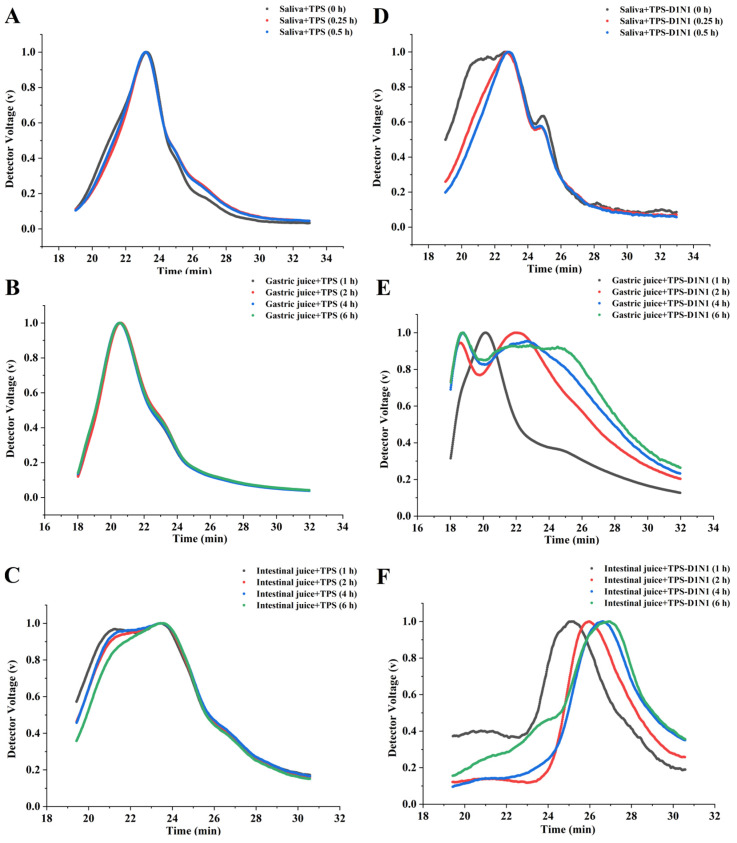
Changes in (**A**–**C**) TPS and (**D**–**F**) TPS-D1N1 MWs during in vitro simulated orogastrointestinal digestion.

**Figure 2 foods-13-01561-f002:**
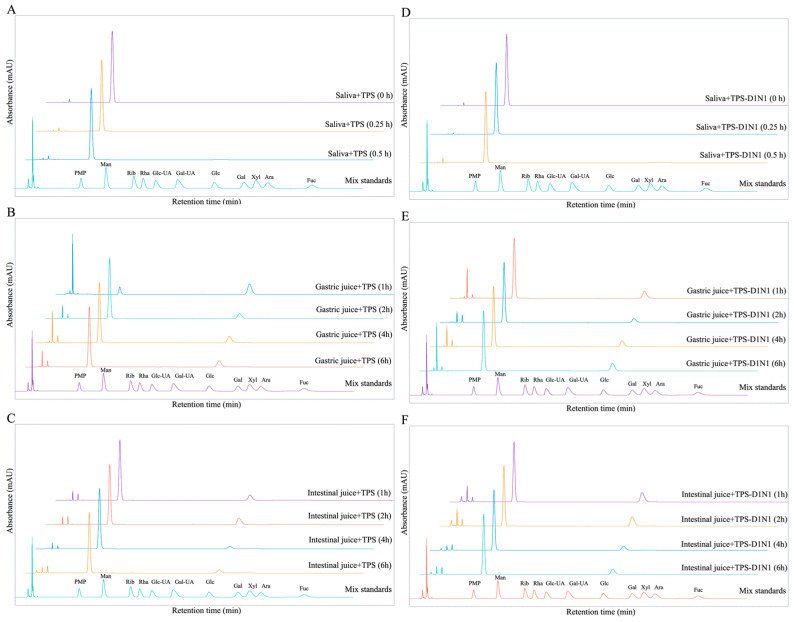
Changes in (**A**–**C**) TPS and (**D**–**F**) TPS-D1N1 monosaccharide content during in vitro simulated orogastrointestinal digestion.

**Figure 3 foods-13-01561-f003:**
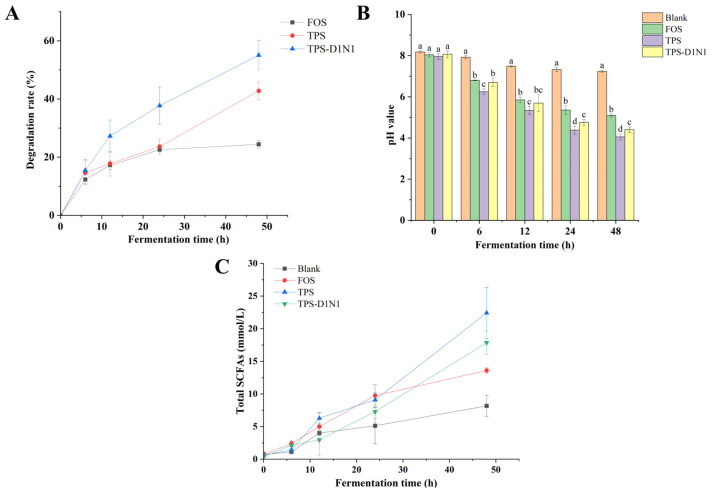
Changes in (**A**) total carbohydrate consumption, (**B**) pH, and (**C**) SCFA production in anaerobic fermentation of TPSs and TPS-D1N1. Values with the same letter are nonsignificantly different (*p* > 0.05), whereas those with different letters are significantly different (*p* < 0.05).

**Figure 4 foods-13-01561-f004:**
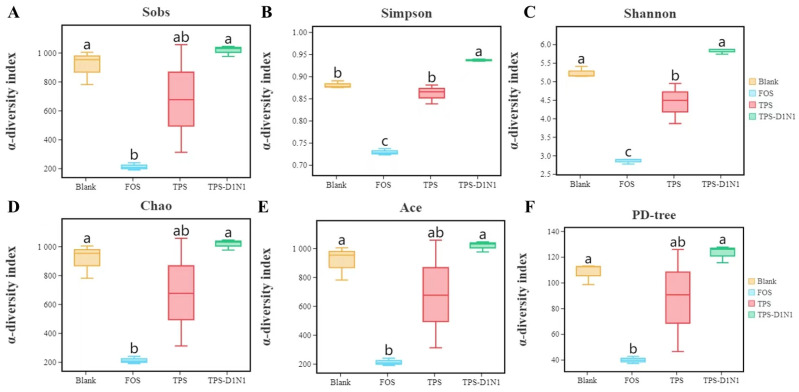
Alpha-diversity of gut microbiota in blank, TPS, TPS-D1N1, and FOS groups (**A**–**F**). Values with the same letter are nonsignificantly different (*p* > 0.05), whereas those with different letters are significantly different (*p* < 0.05).

**Figure 5 foods-13-01561-f005:**
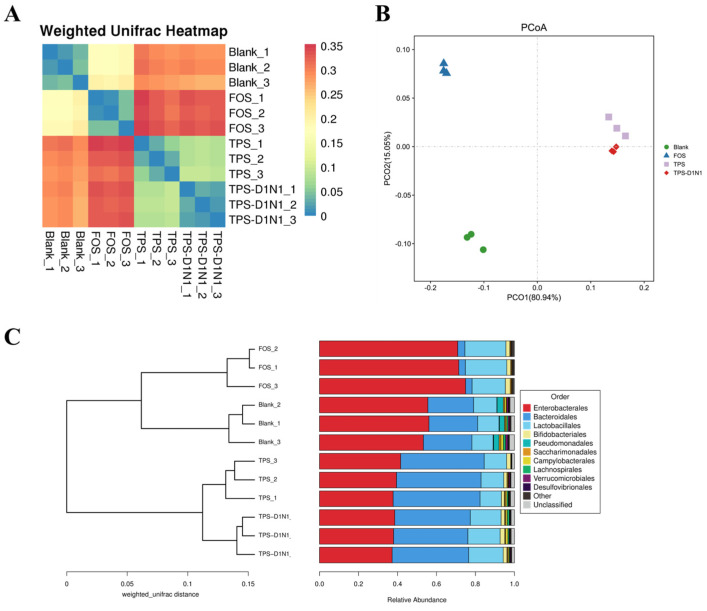
Beta diversity of gut microbiota in blank, TPS, TPS-D1N1, and FOS groups. (**A**) Heatmap. (**B**) PCoA plot. (**C**) Clustering tree based on the weighted UniFrac distances.

**Figure 6 foods-13-01561-f006:**
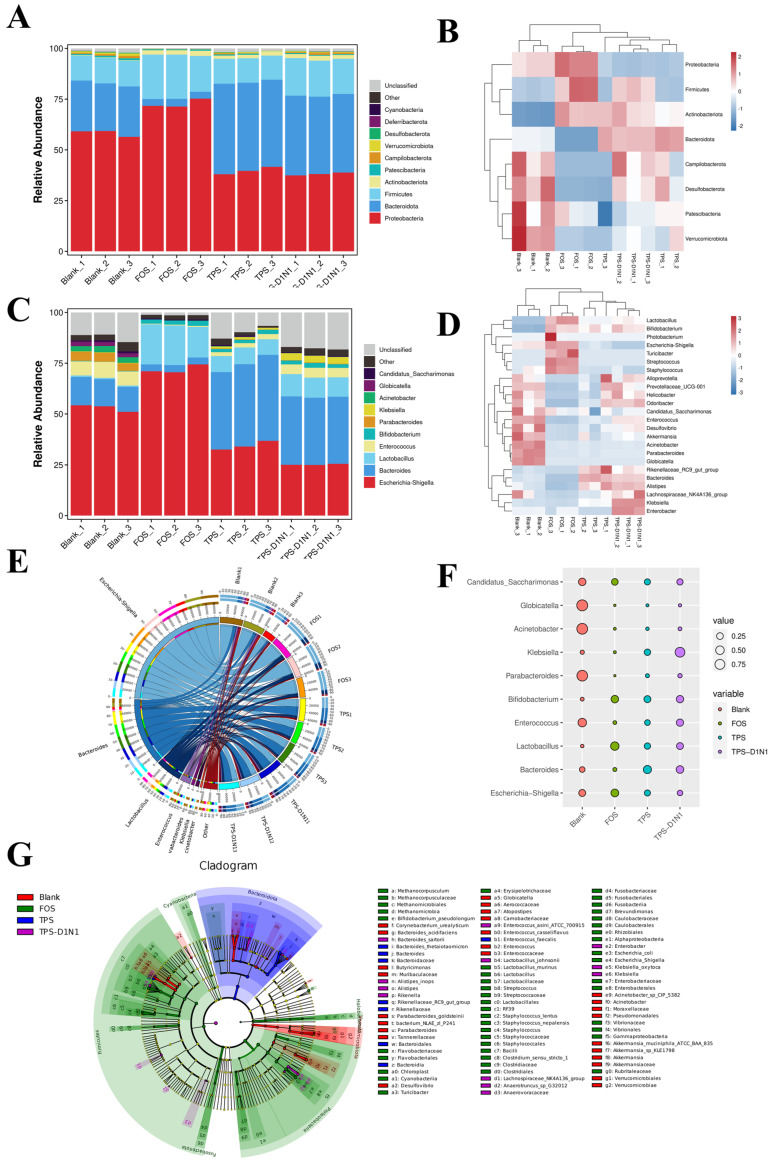
Relative abundance of the bacterial community. (**A**) Percentages and (**B**) heatmap of bacteria at the phylum level. (**C**) Percentages and (**D**) heatmap of species at the genus level. (**E**) Horizontal Circos diagram. (**F**) Horizontal grouping biomarker bubble chart. (**G**) Linear discriminant analysis effect size chart (LDA Score > 2.0).

**Table 1 foods-13-01561-t001:** TPS and TPS-D1N1 characterization.

Parameter	Type	Value	
		TPSs	TPS-D1N1
Main components (%, *w*/*w*)	Carbohydrate	68.89	70.58
	Protein	25.67	11.57 ± 0.10
Monosaccharide content (μg/mg)	Man	9.07	17.56
	Rib	ND	ND
	Rha	26.93	19.26
	Glc-UA	11.80	8.72
	Gal-UA	4.21	47.58
	Glc	18.43	27.72
	Gal	98.89	94.07
	Xyl	4.57	4.15
	Ara	74.65	78.79
	Fuc	2.24	3.19
MW (kDa)	/	478.75	289.10

ND: not detected.

**Table 2 foods-13-01561-t002:** Reducing sugar content in TPS and TPS-D1N1 after orogastrointestinal digestion.

Samples	Reducing Sugar Content (μg/mL)	
	TPSs	TPS-D1N1
Salivary digestion		
0 min	ND	ND
15 min	ND	ND
30 min	ND	ND
Gastric digestion		
1 h	127.08 ± 12.5 ^b^	547.92 ± 38.19 ^b^
2 h	139.58 ± 33.07 ^b^	589.58 ± 29.17 ^ab^
4 h	157.64 ± 6.36 ^ab^	624.31 ± 27.74 ^a^
6 h	182.64 ± 21.38 ^a^	647.92 ± 23.2 ^a^
Small intestinal digestion		
1 h	200.69 ± 46.83 ^b^	681.94 ± 10.49 ^a^
2 h	235.42 ± 41.04 ^b^	673.61 ± 8.67 ^a^
4 h	246.53 ± 45.71 ^b^	729.17 ± 15.02 ^b^
6 h	382.64 ± 27.74 ^a^	770.83 ± 7.22 ^c^

Data are presented as means ± SDs; n = 3. ^a–c^ Means in the same column and within the digestion stage with different superscripted letters differ (*p* < 0.05) according to the Tukey test.

## Data Availability

The original contributions presented in the study are included in the article, further inquiries can be directed to the corresponding authors.

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
