# Peer review of "In Vitro Characterization of Polysaccharides from Fresh Tea Leaves in Simulated Gastrointestinal Digestion and Gut Microbiome Fermentation"

_foods, 2024, doi:10.3390/foods13101561_

Round 1

Reviewer 1 Report

Comments and Suggestions for Authors

The authors describe an interesting use of TPS from fresh tea leaves. I have some minor comments and would like to make them:

1) What do the authors explain the decrease in MW TPS D1M1 after 6h of gastric digestion?

2) Similarly, please explain more fully the increase in TPS reducing sugar over 6h of gastric digestion,

3) The effect of inulin on the intestinal microenvironment is still debated; there are reports claiming that it stimulates the growth of fecal bacteria Clostridium sp;

4) It may be worth mentioning the cross-feeding phenomenon, which in the case of SCFAs is most often from acetate to butyrate; have valerian and caproic acid also been determined?

5) Elevated levels of Firmicutes bacteria are also often found in patients on high-fat diet and obese; their increase in the present study can also be discussed in the context of these reports;

6) The decrease in the diversity of the intestinal microbiota caused by TPS seems problematic. This is often an unfavorable phenomenon. Please comment more on this issue.

Author Response

We feel great thanks for your professional review work on our article. As you are concerned, there are several problems that need to be addressed. According to your nice suggestions, we have made extensive corrections to our previous draft, the detailed response to your comments are listed below.

Our response to your comments is as follows.

  1.  Thank you for pointing this out, and we have revised and supplemented it. In the revised manuscript this change can be found – page 6, and line 266.

The specific reasons are as follows:

At present, it was publically accepted that the polysaccharide tend to form aggregates in a aqueous system. The disruption of aggregates and covalent bond cleavage in polymer chains all can contribute to the molecular weight reduction of polysaccharides. However, the breakdown of glycosidic bonds could result in the increase of the number of the reducing ends. Therefore, the DNS method was used to measure the amount of reducing sugar in this experiment to determine whether the molecular weight change of TPS during digestion was caused by covalent bond breakage. In the results of reducing sugar in the present experiment, we found that the content of reducing sugar increased after 6 h of gastric digestion, indicating that the reduction of TPS molecular weight during gastric digestion was related to the destruction of glycosidic bonds. Studies have shown that polysaccharides can be partially degraded under acidic conditions , so we speculate that the decrease in Mw of TPS after gastric digestion for 6h is related to the acidic environment of the stomach leading to the breakage of glycosidic bonds in the polymer chain.

In the manuscript, we have made appropriate supplement to the reasons for the reduction of MW of TPS after 6h of gastric digestion. Thank you for your advice!

  1. Thank you very much for your comments. We have commented more in the manuscript on the reasons for the increase in TPS reducing sugars during the 6 h gastric digestion. In the revised manuscript this change can be found – page 8, and line 290.

Combined with the data on reducing sugar and monosaccharide composition, it was suggested that the increase in TPS reducing sugar within 6 h of gastric juice digestion was related to the cleavage of glycosidic bonds.  The results of this study showed that the reducing sugar content of TPS increased after 6h of digestion with gastric juice, which indicated that the aggregates in TPS were destroyed and the glycosidic bonds were broken, leading to the increase of reducing sugar content.  The results of monosaccharide composition also showed that a small amount of glucose was produced in gastric juice after digestion, and further suggested that the increase of reducing sugar content may be related to the cleavage of glycosidic bonds.

We have given a more detailed explanation in the manuscript, and thank you again for your suggestions.  Thank you.

  1. Thank you very much for your suggestions. Your suggestions provide a good help to the improvement of this manuscript. Since inulin has been proved to have the potential of prebiotics by most scholars, we use inulin as one of the examples in our results and analysis. Your comments are very helpful. We have reviewed the relevant literature and found that it has been reported that in mice with abnormal metabolism, food rich in inulin causes obvious flora imbalance, resulting in a significant increase in Clostridia and Proteobacteria [1], which has potential safety risks. To avoid controversy, we revised the manuscript to not elaborate on inulin as an illustrative example.

[1] Vishal Singh et al., (2018), Dysregulated Microbial Fermentation of Soluble Fiber Induces Cholestatic Liver Cancer, Cell

  1. Thank you very much for your advice! We measured six SCFAs, It includes acetic acid, propionic acid, butyric acid, isobutyric acid, valeric acid, isovaleric acid, hexanoic acid. Acetic acid, propionic acid and n-butyric acid were the main fermentation products, but valerian and caproic acid were not detected.

I am very sorry for not explaining the category of SCFAs standard in the material, we have supplemented it in the material, thank you!

  1. Thank you very much for your suggestion. In the revised manuscript this change can be found – page 13, and line 448.

We have supplemented the manuscript as follows:

“However, it has been reported that an increased ratio of Firmicutes to Bacteroidetes may contribute to obesity because it is associated with increased colonic fermentation and energy extraction of SCFAs. Moreover, the presence of Firmicutes was found to be associ-ated with diets high in sugar and fat in mice. Therefore, we speculate that there is a close relationship between the increased content of Firmicutes and the intervention of TPS.”

  1. Thank you very much for your advice. We have supplemented and revised the manuscript. The TPS-induced reduction in gut microbiota diversity may be explained by the competitive effects of dominant microbiota.

Further comments on this issue are provided in the manuscript (page 11, line 384 and page 13 line 420), which shows that TPS treatment can effectively regulate intestinal microbiota and enrich Bacteroides.

We tried our best to improve the manuscript and made some changes marked in red in revised paper which will not influence the content and framework of the paper. We appreciate for Editors/Reviewers’ warm work earnestly, and hope the correction will meet with approval. Once again, thank you very much for your comments and suggestions.

Reviewer 2 Report

Comments and Suggestions for Authors

The authors investigated the characteristics of tea polysaccharides during in vitro simulated gastrointestinal digestion and colon fermentation.  The results are valuable, and the design of experiment and applied statistics appropriate.

The main question addressed by the research is the impact of in vitro simulated gastrointestinal digestion on tea polysaccharides (TPSs) and their purified component (TPS-D1N1), as well as their effects on gut microbiome composition, diversity, and short-chain fatty acid (SCFA) production.

This study addresses a gap in the field by investigating the structural stability and digestive behavior of TPSs, particularly those extracted from fresh tea leaves, which have been less studied compared to finished tea products. It also explores the impact of TPSs on gut microbiota composition, which is a relatively novel area of research. The study's focus on the interaction between TPSs and gut microbiota is significant given the increasing interest in the gut microbiome and its influence on human health. The investigation of polysaccharides' effects on gut microbiota, along with their structural changes during digestion, is original and relevant for understanding the potential health benefits of TPSs.

Contribution compared with other published material: The study's findings contribute to understanding how TPSs can modulate gut microbiota and potentially improve host health.

While the study provides valuable insights, improvements in methodology could enhance its robustness. For example, conducting future experiments with human subjects or animal models could validate the findings from in vitro models and provide a more comprehensive understanding of TPSs' effects.

The conclusions drawn from the evidence presented are consistent.

The references provided are appropriate and cover relevant areas.

I suggest authors to change Abstract sentence “Then, we investigated the characteristics of TPSs during gastrointestinal digestion and colon fermentation using an in vitro simulated digestion and fermentation model;  moreover, the effects of 15 TPSs on gut microbiota were explored” into “Then, we investigated the characteristics of TPSs during an in vitro simulated digestion and fermentation;  moreover, the effects of 15 TPSs on gut microbiota were explored”, to avoid duplication of the same words.

Also, Figure 3A and 3B, there is word “fermentation” on graphs, which should be corrected into “fermentation”.

Author Response

Thanks very much for taking your time to review this manuscript. I really appreciate all your comments and suggestions! We believe that your suggestions will be of great help to our future research work ! In the future work, we will continue to conduct in vivo study of TPSs to confirm its biological activity and mechanism of action. We have amended the paper following the indications that you and the referees gave us. Please find my itemized responses in below and my revisions/corrections in the re-submitted files.

Response 1: Thank you for your suggestion, we’ve rectified. The Abstract sections have been improved.

Response 2: We feel sorry for our carelessness. In our resubmitted manuscript, the typo is revised. Thanks for your correction.

We appreciate the thoughtful review and constructive feedback provided by the reviewers. We appreciate for Editors/Reviewers’ warm work earnestly, and hope the correction will meet with approval. Once again, thank you very much for your comments and suggestions.
